# Ribosomal DNA Sequence-Based Taxonomy and Antimicrobial Activity of *Prorocentrum* spp. (Dinophyceae) from Mauritius Coastal Waters, South-West Indian Ocean

**DOI:** 10.3390/md21040216

**Published:** 2023-03-28

**Authors:** Prakash Mussai, Jacob Larsen, Abdulwahed Fahad Alrefaei, Rajesh Jeewon

**Affiliations:** 1Department of Health Sciences, Faculty of Medicine and Health Sciences, University of Mauritius, Reduit 80837, Mauritius; 2IOC Science and Communication Centre on Harmful Algae, Biological Institute, University of Copenhagen, Universitetsparken 4, DK-2100 Copenhagen, Denmark; 3Department of Zoology, College of Science, King Saud University, P.O. Box 2455, Riyadh 11451, Saudi Arabia

**Keywords:** dinoflagellates, *Prorocentrum fukuyoi* complex, *Prorocentrum rhathymum*, *Prorocentrum lima* complex, morphology, phylogeny, antimicrobial activity, Mauritius

## Abstract

Microalgae are unicellular organisms and commonly present in the euphotic zone of marine ecosystems. From the western coast of Mauritius, three strains of *Prorocentrum* species were isolated from macrophytes and cultured under standard laboratory conditions. Morphologies were examined by light, fluorescence, and scanning electron microscopy, and phylogenetic analyses were based on partial large subunit LSU rDNA (D1-D2) and ITS1-5.8S-ITS2 (ITS) regions. Three *Prorocentrum* species, including the *P. fukuyoi* complex, *P. rhathymum*, and *P. lima* complex, were identified. The antimicrobial activities were assayed against potential human pathogenic bacterial strains. The highest zone of inhibition was recorded for intracellular and extracellular protein extracts of *Prorocentrum rhathymum* against *Vibrio parahaemolyticus*. The polysaccharide extracts of the *Prorocentrum fukuyoi* complex had a higher zone of inhibition (24 ± 0.4 mm) against MRSA at a minimum concentration of 0.625 μg/mL. The extracts from the three *Prorocentrum* species had different levels of activity against the pathogens used, and this can be of scientific interest in the search for antibiotics from natural marine sources.

## 1. Introduction

Microalgae are a highly diverse group of unicellular organisms, ubiquitous in the euphotic zone of marine ecosystems, and form the basis of the marine food chain. They are known to produce a variety of polymeric substances of potential interest to the pharmaceutical industry [1,2]. Dinoflagellates are present within a myriad of microalgae inhabiting benthic substrates such as algal turf, seagrasses, macroalgae, rocks, and sediments [3]. They are mostly recognized for their negative impact resulting in harmful algal bloom (HAB) [4]. Though most of the studies have been devoted to elucidating dinoflagellate toxins owing to illnesses such as ciguatera poisoning, paralytic shellfish poisoning, and neurotoxic shellfish poisoning [5,6,7], they also possess unique bio-toxins and biologically active substances [8]. Hence, dinoflagellates are likely candidates in the development of new efficient antimicrobial drugs, especially given the emergence of multi-drug-resistant bacteria. Traditional methods to assess microalgal diversity from coastal environments relied mostly on traditional microscopy methods. However, given the problems arising from phenotypic plasticity and cultural variation, there has been a need to incorporate the use of DNA sequence data. The latter, especially ribosomal DNA, has been widely used, not only across microalgae but among a wide range of microorganisms [9,10,11,12].

The genus *Prorocentrum* C.G. Ehrenberg (1834) contains potentially toxic dinoflagellate species [13,14] and has been reported to possess promising bioactive compounds [15,16,17,18]. So far, 81 *Prorocentrum* species are taxonomically accepted [19], and of these, 35 have been reported as benthic species [20,21,22,23,24]. The aim here is to characterize epiphytic *Prorocentrum* species from Mauritius coastal waters based on morphology and DNA sequence analyses and investigate their antimicrobial properties of crude extracts against selected human pathogenic bacteria.

## 2. Results

### 2.1. Morphological Features of Prorocentrum Species Isolated

#### 2.1.1. *Prorocentrum rhathymum*

Based on the morphologies examined, our isolate ISOP15 is identified as *Prorocentrum rhathymum* as the cells were asymmetrical and oval or ellipsoidal in apical and antapical views (Figure 1), with photosynthetic containing reticulated golden-brown chloroplasts (Figure 1a). The cell length was 20.45–23.82 μm (average = 22.30 ± 1.07 μm, *n* = 30), and its width was 15.03–18.18 μm (average = 16.59 ± 1.00 μm, *n* = 30). Their length-to-width (L/W) ratios were 1.35 ± 0.06. The widest part of the cell was in the middle- or upper part (Figure 1a,b). A depression in the anterior region of the right valve bore the periflagellar collar present as a spine (Figure 1a–c). Posterior pores were radially arranged (Figure 1b,c); also, one apical row of pores was observed at the anterior marginal end (Figure 1c). The plate center was devoid of pores.

#### 2.1.2. *Prorocentrum fukuyoi* Complex

Based on the morphologies examined, our isolate ISOP40 is identified as a *Prorocentrum fukuyoi* complex, given that the cells were symmetrical and the region next to the apical region was flattened rather than rounded (Figure 2a,b). Pyrenoid with ring-shaped starch sheath was absent. The cell length was 17.71–21.78 μm (average = 19.49 ± 1.00 μm), and its width was 15.12–18.09 μm (average = 16.35 ± 0.59 μm). Their length-to-width (L/W) ratios were 1.19 ± 0.04. The cells had a V-shaped indentation, and probably two different pores size classes were present (Figure 2b). Rows of pores radiating towards the center, scattered pores around the periphery of the value and in the center (Figure 2b). There was also a spine or flange near the V-shaped indentation (Figure 2a,b).

#### 2.1.3. *Prorocentrum lima* Complex

Based on the morphologies examined, our isolate P29 is identified as a *Prorocentrum lima* complex as the cells were photosynthetic, containing numerous golden-brown chloroplasts and a central pyrenoid with a thick starch sheath. Cells presented a nearly ovoid shape (Figure 3). An elongated oval nucleus is posteriorly located (Figure 3a). The cell length was 25.19–28.57 μm (average = 26.83 ± 0.77 μm, *n* = 30), and its width was 18.47–21.78 μm (average = 19.70 ± 0.99 μm, *n* = 30) wide. Their length-to-width (L/W) ratios were 1.36 ± 0.04. Cells exhibited a smooth thecal surface with scattered value pores, except in the central area, which was devoid of pores (Figure 4). Thecal pores shapes ranged from oblong to kidney-shaped pores (Figure 4). Located at the periphery of the values was a ring of evenly spaced marginal pores, ovoid to oblong shaped (Figure 4b,c). Cells were observed with smooth intercalary bands (Figure 4b). The periflagellar area was V-shaped on the right thecal plate. Following the nomenclature proposed by Hoppenrath et al. [20], nine periflagellar platelets (1a,1b, 2, 3, 4, 5, 6, 7, and 8) were observed in this species (Figure 4c). The collar of cells in the left value was flattened or a slightly curved collar (Figure 4a,c).

### 2.2. Phylogenetics of Prorocentrum Species Isolated

#### 2.2.1. *Prorocentrum rhathymum*

The rDNA LSU phylogeny (Figure 5) placed the Mauritian isolate ISOP15 in a clade comprising taxa currently assigned to *Prorocentrum rhathymum* species, and hence our strain is identified as *Prorocentrum rhathymum*. *Prorocentrum rhathymum* isolate ISOP15, AY259166 (Heron Island), AJ567468 (Réunion Island), and MH348965 (Parcel Islands), clustered together in a sub-clade with high support (91 BS/1.00 PP). The nucleotide differences for the LSU rDNA (D1-D2) among the sequences assigned *Prorocentrum rhathymum* ranged from 0 to 137 bases in 739 bp, which was equal to a p-distance from 0.000% to 18.538%.

#### 2.2.2. *Prorocentrum fukuyoi* Complex

The rDNA LSU phylogeny (Figure 5) placed the Mauritian isolate ISOP40 in a distinct clade comprising taxa currently assigned to the *Prorocentrum fukuyoi* complex. *Prorocentrum fukuyoi* isolate ISOP40, DQ336192 (Fiji), and type strain DQ336191 (Australia) clustered together in a sub-clade with support (87 BS/1.00 PP). They were closely related to another sub-clade comprising JX912183 (Germany), HQ414227 (Spain), and JX912173 (France). The nucleotide differences for the LSU rDNA (D1-D2) among the 23 sequences assigned *P. fukuyoi* complex ranged from 1 to 158 bases within the 739 bp sequenced, which was equal to a p-distance from 0.135% to 21.380%.

#### 2.2.3. *Prorocentrum lima* Complex

The rDNA ITS phylogeny (Figure 6) placed the Mauritian isolate P29 in a distinct subclade with high bootstrap and probability values, together with AB189742 (Tahiti) and AB189771 (Japan), and as part of a larger clade A related to *P. lima* species complex. The *Prorocentrum lima* complex from clade A was closely related to the *Prorocentrum lima* complex from clade B (AB189746, EU927506, AB189747, KU722938, and KJ934790) and *Prorocentrum arenarium* (KM266622, EU244470, KM266623, and KM266619). *Prorocentrum lima* complex (Clades A and B), as well as *Prorocentrum arenarium*, grouped together with a strong support (100 BS/1.00 PP) and were sisters to the *P. lima* complex from clade C, *P. porosum*, *P. caipirignum*, and *P. hoffmannianum*. The nucleotide differences for the ITS1-5.8S-ITS2 (ITS) regions among the 13 sequences assigned *P. lima* complex from Clade A ranged from 1 to 22 bases within the 566 bp sequenced, which was equal to a p-distance from 0.176% to 3.886%.

### 2.3. Antimicrobial Activities of Extracts

The three species were screened against a set of bacterial strains composed of both Gram-negative and Gram-positive bacteria. All the extracts were standardized at 0.2 mg/mL for each assay.

In the qualitative evaluation of the antibacterial activity (Table 1), the intracellular extract of *P. rhathymum* showed the biggest inhibition zone in *V. parahaemolyticus* (G−) (30 ± 0.0 mm). While *P. lima* complex showed the strongest activity against *S. typhimurium* (G−) (27 ± 0.4 mm) and *E. faecalis* (G+) (25 ± 0.4 mm). Similarly, the *P. fukuyoi* complex had the highest activity in two Gram-negative with 21 ± 0.4 mm. Extracellular protein extract of *P. rhathymum*, *P. lima* complex, and *P. fukuyoi* complex showed the biggest inhibition zone in *V. parahaemolyticus* (G−) with 27 ± 0.5 mm, 18 ± 0.5 mm, and 20 ± 0.4 mm, respectively. Likewise, extracellular polysaccharides of *P. rhathymum* and *P. lima* complex showed the strongest activity in *S. typhimurium* (G−) with 28 ± 0.2 mm and 0.6 ± 0.2 mm, respectively. In comparison, the *P. fukuyoi* complex against Gram-negative Methicillin-resistant *Staphylococcus aureus* (MRSA) showed 24 ± 0.4 mm. The negative control produced no observable zones.

In the quantitative determination of the antibacterial activity (Table 2), intracellular extracts of *P. rhathymum* and *P. lima* complex showed the lowest MIC values against *S. pneumonia* at 1.25 μg/mL and 0.625 μg/mL, respectively. In comparison, the *P. fukuyoi* complex showed the lowest MIC value against *V. parahaemolyticus* at 0.625 μg/mL. Extracellular protein extracts of *P. rhathymum* and *P. lima* complex showed the lowest MIC values against *P. mirabilis* and *S. pneumonia* at 0.625 μg/mL and 1 μg/mL, respectively. On the other hand, extracellular polysaccharide extracts of *P. rhathymum* and *P. fukuyoi* complex showed the lowest MIC values against *V. parahaemolyticus* and *S. pneumonia* Methicillin-resistant *Staphylococcus aureus* (MRSA) at 0.625 μg/mL.

Overall, the intracellular extracts, extracellular protein, and polysaccharides extracts of the *Prorocentrum* isolates had significantly varying degrees of antibacterial potential (*p* < 0.05) against the selected Gram-negative and Gram-positive bacteria.

## 3. Discussion

In Mauritius, investigations on *Prorocentrum* species have primarily been as part of a national monitoring program for unraveling potentially harmful benthic dinoflagellates [25,26,27]. *Prorocentrum* species are recognized to be morphologically similar with apparent plasticity [20,21,28,29], and their identification has been rather difficult based on morphology alone. Hence, there is a necessity to supplement morphological observations with DNA sequence-based analyses to properly identify isolates, at least at the species level.

The *Prorocentrum* isolate (ISOP15) established during the present study matched the oval or ellipsoidal shape in the apical and antapical views previously ascribed to *P. rhathymum* species. Under light and fluorescence microscopy, the main morphological traits attributed to the pore pattern (one apical row, posterior radial rows, and central plate devoid of pores) and the presence of a spine in the right value of *P. rhathymum* species were observed. However, cells of isolate ISOP15 were characterized by smaller sizes as compared to other existing strains (20.45–23.82 μm long and 15.03–18.18 μm wide) as compared to 32–29 μm long and 20–22 wide μm [30], 38–40 μm long and 22–25 wide μm [31], 28–33 μm long and 18–23 wide μm [32], 31.4–33.3 μm long and 25.5–30.4 wide μm [33], 29–34 μm long and, 29.5–35.0 μm long and 20.8–26.0 μm wide [34]. The phylogenetic analyses of the LSU rDNA (D1⁄D2) region showed the intermixing of *Prorocentrum rhathymum* and *Prorocentrum mexicanum* strains. As outlined by Gómez et al. [35], sequences prior to 2003 were submitted as *P. mexicanum*, while sequences after 2003 were submitted as *P. rhathymum*, and very frequently, both names have been used interchangeably. The authors further proposed *P. rhathymum* as a junior synonym of *P. mexicanum*. In this study, the identity of isolate ISOP15 is established as *Prorocentrum rhathymum*.

*Prorocentrum* isolate (ISOP40), isolated during this study, presents close affinity to the type species of *P. fukuyoi* collected from Port Botany, Sydney, Australia. However, ISOP40 cells were of lower mean size (19.49 ± 1.00 μm long and 16.35 ± 0.59 μm wide) as compared to the original description of *Prorocentrum fukuyoi* (38 ± 3.2 μm long and 26 ± 5.5 μm wide). Similarly, ISOP40 cells were of a lower length range compared to those described by Faust [36] as *Prorocentrum emarginatum* (35–40 μm)—which may likely be considered as *P. fukuyoi* according to Chomérat et al. [37] and to *Prorocentrum* cf. *emarginatum* described by Luo et al. [38] with a mean size of 37.0 ± 1.5 μm long and 32.9 ± 1.8 μm wide. The original descriptions with regards to the size range given for *Prorocentrum emarginatum* by Fukuyo [31] were 35–36 μm long and 32 μm wide. This is indicative that size range cannot be used to distinguish between these closely related species. In this study, the phylogenetic analyses of LSU rDNA (D1⁄D2) regions showed the intermixing of strains reported under *Prorocentrum fukuyoi*, *Prorocentrum emarginatum*, and *Prorocentrum emarginatum/fukuyoi* and recovered four clades in agreement with Wu et al. [39]. Our result indicated that ISOP40 fell within the clade from the type material. The results supported the identity of our isolate ISOP40 as *Prorocentrum fukuyoi* complex.

*Prorocentrum lima* is largely distributed from temperate to tropical marine areas worldwide [20,28,31]. Past studies indicated the cell shape, as expressed from the length/width ratio, of *P. lima* species as widely variable with ovoid, ellipsoid-oblong, ovate-oblong (oval-oblong), ovate-piriform, and round [13,20,21,23,28,33,37,38,40,41,42,43,44,45]. Though early observations suggested that morphological gradations in cell size and shape within and between sites were minimal and not significant for *P. lima* [40,46]; however, it is now regarded as a “*P. lima* complex” [20,28,33]. As discussed in Nascimento, Mendes, Menezes, Rodríguez, Alves-de-Souza, Branco, Riobó, Franco, Nunes, Huk, Morris, and Fraga [22], the morphological variability in *Prorocentrum lima* may not be as wide as earlier reported, and more than one species have been incorporated in *P. lima* species complex. Lately, based on morphological and genetic analyses, *Prorocentrum caipirignum* and *P. porosum* [22,47] have been described from this species complex. The *Prorocentrum* isolate (P29) established during the present study matched the ovoid shape currently ascribed to the *P. lima* species complex [20,28], but the periflagellar region was composed of nine platelets. Maximum likelihood analysis and Bayesian inference based on the ITS regions generated in this study agree with previous reports [13,21,28,48] in recognizing several clades from *Prorocentrum lima* species complex and related species. In the phylogeny, *P. lima* complex (P29) is reported in Clade A, which also includes isolates from broader geographic locations.

Though there are few studies of marine natural products from micro-algae [17], nevertheless bioprospecting studies for new marine bioactive compounds, especially for antioxidant, anti-inflammatory, anticancer, anti-diabetes, and antibacterial activities, are gaining strength [15,17,49,50]. A recent bioprospecting study on 33 microalgae strains [17] for their antimicrobial, antiproliferative, and apoptotic potential showed promising results for a few species from the genus *Prorocentrum*, namely; *P. arenarium*, *P. hoffmannianum*, and *P. reticulatum.* Data reported here showed that intracellular extracts of the *P. lima* complex were effective against all Gram + ve and Gram − ve bacterial strains, especially against *E. faecalis* (G+) (25 ± 0.4 mm) and *S. typhimurium* (G−) (27 ± 0.0 mm) (Table 1). However, it must be emphasized that the strongest antibacterial effect was from intracellular extracts of *Prorocentrum rhathymum* with a significant inhibition zone for *V. parahaemolyticus* (G+) (30 ± 0.0 mm) at a minimum concentration of 2.5 μg/mL (Table 2). Overall, data reported here showed that toxins extracted from the three cultured *Prorocentrum* species had significant inhibitory properties against both Gram-positive and Gram-negative bacteria.

## 4. Materials and Methods

### 4.1. Study Area and Sample Collections

Macrophyte samples, including macroalgae and seagrass, namely *Turbinaria* sp., *Padina* sp., *Dictyopteris* sp., *Syringodium* sp., and *Halodule* sp., were collected from Flic-en-Flac (20°16′22″ S, 57°22′14″ E), Wolmar (20°18′39″ S, 57°21′56″ E) and Trou aux Biches (20°2′24″ S, 57°32′25″ E) in September 2015 (Figure 7). Macrophyte samples were collected in 800 mL wide-mouthed plastic jars, filled with local seawater to approximately one-third, and kept at dark ambient temperature. In the laboratory, samples were vigorously shaken for approximately one min and were sieved through 300, 125, and 20 μm filters. Single cells of *Prorocentrum* spp. were isolated, from a mixed assemblage of dinoflagellates, including *Coolia* spp., *Ostreopsis* spp., and *Gambierdiscus* spp., using the micropipette technique [51] under a compound microscope. The *Prorocentrum* cells were placed in individual wells of a 24-well plate filled with filter-sterilized f/2 medium [52] and incubated at 26 °C under an irradiance of 50 μmoL photons m^−2^ s^−1^ on a 14 h:10 h of light/dark cycle. Cell growth was monitored, and when sufficient cell density was achieved, surviving isolates were transferred to 125 mL polycarbonate Erlenmeyer flasks containing 80–100 mL, half-strength culture medium. Established cultures were sequentially transferred to full strength medium and maintained at 26 °C, an irradiance of 100 μmoL photons m^−2^ s^−1^ in a 14 h:10 h of light/dark cycle, without suspension. Three *Prorocentrum* monoclonal cultures were established and used in this investigation: isolate P29 from Flic-en-Flac, isolate ISOP15 from Wolmar, and isolate ISOP40 from Trou aux Biches.

### 4.2. Light Microscopy (LM) and Scanning Electron Microscopy (SEM) Observations

For LM observations, cells were observed using a compound microscope (Leica DM1000) equipped with a UV epifluorescence system. The shape and position of the nucleus was achieved by fixing 1 mL exponential cultures with 3.7% formaldehyde (15 min). Cell size was estimated from the length and width data of 30 cells for each strain, determined at ×1000 magnification. For epifluorescence observations, visualization of cellulose thecal plates was performed on cells stained with Fluorescent Brightener 28 (Calcofluor White M2R) (0.2 μg mL^−1^) [53]. For scanning electron microscopy, samples were prepared according to the same procedures as our previous study on *Gambierdiscus caribaeus* [54].

### 4.3. DNA Extraction and PCR Amplification

DNA extraction for the isolates was conducted as follows: 5–10 mL of clonal cultures were centrifuged, and genomic DNA was extracted from the cell pellets using the 10% Chelex^®^ 100 [55]. PCR amplification for 25 μL reactions was performed using OneTaq^®^ 2X Master Mix (New England Biolabs, Hitchin, UK), following manufacturers’ instructions with primer pair D1R-F & D2C-R for the LSU region [56], and primer pair ProITS-F–ProITS-R [28] for the ITS regions. The PCR reactions were carried in a thermocycler (Applied Biosystems, Foster City, CA, USA), and the amplification profile for the LSU region was set as the initial denaturation step at 95 °C for 5 min, followed by 35 cycles at 95 °C for 1 min; annealing at 52 °C for 1 min; extension at 72 °C for 2 min, with a final extension at 72 °C for 7 min. The amplification profile for the ITS regions were set as denaturation at 94 °C for 1 min, annealing at 64.4 °C for 1 min, and extension at 72 °C for 10 min. This cycle was repeated 30 times and followed by a final extension step at 72 °C for 10 min. DNA sequencing was performed using the same primers at a commercial service (Inqaba, Pretoria, South Africa). The LSU and ITS sequences of the *Prorocentrum* isolates obtained in this study have been deposited in Genbank with the following accession numbers: ISOP29 (OQ352631); ISOP40 (OQ352626); ISOP15 (OQ352630).

### 4.4. Phylogenetic Analyses

The new sequences of partial LSU and ITS rDNA were aligned with those of other *Prorocentrum* rDNA sequences downloaded from the Genbank database. Multiple sequence alignment was performed in the Geneious Prime 2023.0.1 (Biomatters Ltd., Auckland, New Zealand) [57] using the Clustal Omega 1.2.2 plug-in. Phylogenetic model selection for Maximum Likelihood (ML) method was performed on MEGA X software, version 10.1.7 [58]. A Tamura-Nei (TN93 + G, G = 0.7) model was selected in LSU, while in the ITS phylogeny, a Tamura-Nei (TN93 + G + I, G = 2.3) model was used. Maximum Likelihood analyses were conducted in MEGA X software. The optimal ML tree was subjected to 1000 bootstrap replicates. The phylogenetic relationships were also determined using Bayesian Inference (BI) method and, in this case, performed in the Geneious Prime platform via Mr Bayes 3.2.6 plugin [59]. The best substitution models were obtained across the entire GTR model space. Markov Chain Monte Carlo procedure was based on two independent trials with four chains each. Each chain was run for 2,000,000 generations, and trees were sampled every 1000 trees. A 50% majority-rule consensus tree was drawn from the last 1000 trees. All final spilt frequencies were <0.01. The phylogenetic trees shown herein to depict the relationships of species are derived from the ML analyses and supported by posterior probability values from the bayesian analysis.

### 4.5. Antimicrobial Potential

Antimicrobial activities of crude extracts from the three *Prorocentrum* isolates were assayed against pathogenic bacteria. This was conducted using the disc diffusion method [60], and the minimum inhibitory concentration was determined using the method of the Clinical and Laboratory Standards Institute, M0\7-09 [61].

#### 4.5.1. Intracellular Extract

Cells were harvested when the cultures reached the stationary phase (approx. 25 days) by centrifugation at 4000 rpm for 15 min at room temperature. Following centrifugation, the supernatant (i.e., extracellular extract) was collected, and the pellets were used for intracellular extraction as it is for toxin extraction [62]. The cell pellets were extracted by adding 5 mL of methanol/water (1:1, *v*/*v*) with 0.2% acetic acid solution and sonicated for 3 min in pulse mode while cooling the solution with an ice bath. The mixture was centrifuged at 5000 rpm at 4 °C for 10 min, and the resulting supernatants were transferred to clean centrifuge tubes. The pellets were resuspended in the same solvent and extracted twice following the same procedure, and the supernatants were combined, giving a total volume of 15 mL. The intracellular extracts were dried until the solvent was completely removed. The dry weight of the residues was recorded. Finally, the residues were resuspended in dimethyl sulfoxide (DMSO) to a concentration of 0.2 mg/mL and stored at −20 °C.

#### 4.5.2. Extracellular Proteins and Polysaccharides Extracts

Extracellular proteins were precipitated directly from the growth medium, according to Sánchez et al. [63]. The supernatants were filtered (0.45 μm filter), and to the filtrate, ice-cold trichloroacetic acid (TCA—6% *w*/*v*) was added. The mixture was stirred and incubated for 2 h at 4 °C. The TCA precipitates were recovered by centrifugation at 10,000 rpm for 15 min at 4 °C. The clear supernatants were collected for extracellular polysaccharides analysis. The resulting pellets were washed with chilled acetone (90%) and allowed to dry at room temperature. Extracellular polysaccharides were precipitated by adding 3 volumes of cold absolute ethanol to the TCA supernatant and incubated for 24 h at 4 °C. The precipitates were then centrifuged at 10,000 rpm for 15 min at 4 °C and dried overnight at 40 °C.

#### 4.5.3. Bacterial Pathogens

The above extracts were checked for their ability to inhibit the growth of six human–pathogenic bacteria, namely: Gram-positive Methicillin-resistant *Staphylococcus aureus* (MRSA), *Enterococcus faecalis* ATCC 29212, *Streptococcus pneumoniae* ATCC 49619 and Gram-negative *Proteus mirabilis* strain NCTC 11938, *Vibrio parahaemolyticus* ATCC 17802, *Salmonella typhimurium* ATCC 14028. The bacterial species were grown as follows: MRSA, *Vibrio parahaemolyticus*, *Salmonella typhimurium*—Cystine Lactose Electrolyte Deficient (CLED) agar for 24 h at 37 ± 2 °C. *Enterococcus faecalis* and *Proteus mirabilis*—Nutrient Broth for 48 h at 37 ± 2 °C, *Streptococcus pneumoniae*—Blood Agar for 24 h at 37 ± 2 °C.

### 4.6. Antimicrobial Assay

#### 4.6.1. Antimicrobial Susceptibility Test

The disc diffusion method was used to test antimicrobial activities. First, 100 µL of the inoculum containing approximately 10^7^ CFU/mL of indicator pathogenic bacteria was uniformly spread on Muller-Hinton agar (MHA) plates and air-dried for 5 min. The sterile paper disc of 6 mm in diameter and 1mm thick were placed on the surface of the MHA. Accordingly, an aliquot of 2 µL of *Prorocentrum* extracts (0.2 mg/mL) was applied on the sterile paper discs, positive control (streptomycin—10 µg/mL) and negative control (DMSO) were laid down on the surface of the inoculated plates. All test samples were processed in quintuplicate. After incubation at 37 ± 2 °C for 24 h, evidence of antimicrobial activity was marked by clear zones around the discs. The diameters of the zones of inhibition were then measured (mm).

#### 4.6.2. Determination of Minimum Inhibitory Concentration

The minimum inhibitory concentration (MIC) of *Prorocentrum* extracts for antimicrobial activity was determined using the serial microdilution method. This was performed in disposable, sterile plastic microtitration plates with 96-wells and used extracts that showed positive antimicrobial activity by disc diffusion method. Briefly, the first column of the microtitration tray was filled with 100 µL of the extracts (stock concentration 0.2 mg/mL) and was mixed with 100 µL inoculum of bacteria (approximately 10^7^ CFU/mL) and 100 µL Muller-Hinton broth. Then 100 µL of micropipette mixed amounts are taken from the first column wells and transferred to the second column and so through. The wells of which were filled with an equal volume of inoculum and broth up to a 200 µL volume. Each extract was treated in triplicates. Streptomycin was used as the positive control at the concentration of 10 μg/mL, while the negative control was DMSO. The microtitration plates were sealed in polythene bags and incubated at 37 ± 2 °C for 24 h. p-Iodonitrophenyltetrazolium violet (INT—0.2 mg/mL) was used as an indicator, and the change in color of each well was observed following incubation at 37 °C for 30 min.

#### 4.6.3. Statistical Analysis

Mean ± SD with One-way ANOVA at 5% was calculated for the antimicrobial assays. The statistical analyses of the data were carried out on Statistical Package for Social Scientists (SPSS), Version 20.0, and tabulated using Microsoft Excel.

## 5. Conclusions

In this study, three isolates of *Prorocentrum* were established from the western coast of Mauritius, representing three different species, namely *P. fukuyoi*, *P. rhathymum*, and *P. lima*. This clearly demonstrates that these species are rather cosmopolitan, and there might be more species to be discovered should extensive sampling be done in the coastal regions of Mauritius. To avoid any subjective taxonomic interpretation with respect to the identity of our samples, we provided all morphological details coupled with rDNA sequence-based analyses. Extracts of the three *Prorocentrum* isolates yielded desirable effects against the selected human pathogens, especially *V. parahaemolyticus*, providing preliminary scientific data for the development of potential novel antibiotics from marine sources.

## Figures and Tables

**Figure 1 marinedrugs-21-00216-f001:**
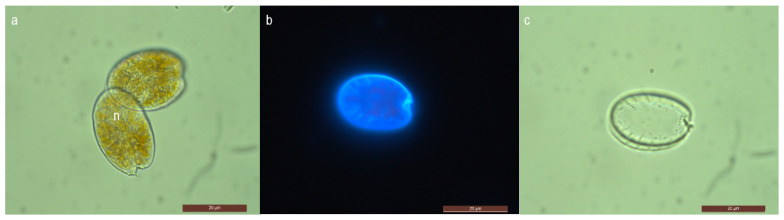
Mauritian isolate of *Prorocentrum rhathymum* (ISOP15); (**a**) light micrographs in the right valve—periflagellar collar appears as a spine, and showing nucleus position (n) in the posterior half of the cells; (**b**) fluorescence micrograph—rows of pores radially arranged and perpendicular to the value margin, scattered pores, the spine, an excavation of periflagellar area in the right value; (**c**) empty valves—rows of radially arranged pores and presence of scattered pores; scale bars (**a**–**c**) = 20 μm.

**Figure 2 marinedrugs-21-00216-f002:**
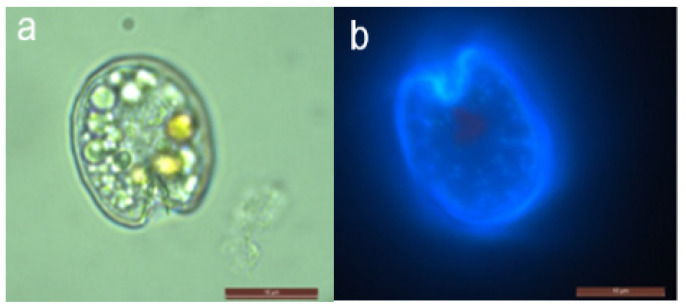
Light/fluorescence micrographs of *Prorocentrum fukuyoi* complex (ISOP40); (**a**) cell with round vesicles; (**b**) fluorescence micrograph—right value, scattered pores around the periphery of the value and center devoid of pores; scale bars (**a**,**b**) = 20 μm.

**Figure 3 marinedrugs-21-00216-f003:**
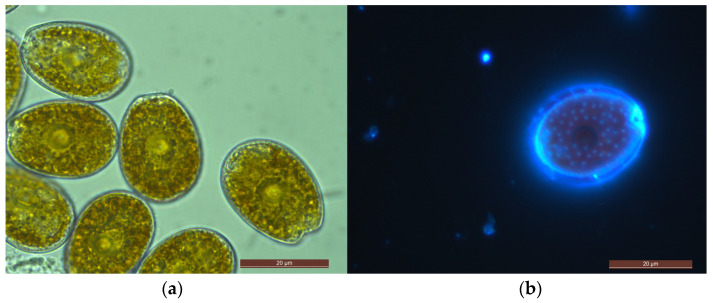
Light and fluorescence micrographs of *Prorocentrum lima* complex (P29). (**a**) LM: right value view showing the central pyrenoid (p) surrounded by a starch sheath and the posterior nucleus (n). (**b**) fluorescence micrograph—right value cell showing the shape, V-shaped flagellar area, and value pores. Scale bars (**a**,**b**) = 20 μm.

**Figure 4 marinedrugs-21-00216-f004:**
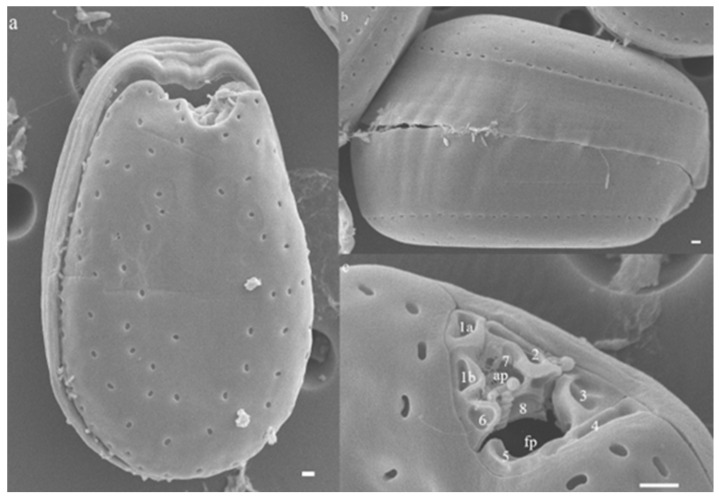
Scanning electron micrograph of *Prorocentrum lima* complex (P29). (**a**) Right valve showing scattered thecal pores. (**b**) An old cell with broad megacytic zones. (**c**) Platelets of the periflagellar area showing night platelets—1a, 1b, 2, 3, 4, 5, 6, 7, and 8. The accessory pore (ap) is composed of platelets 1a, 2, 1b, 7, 8. The flagellar pore (fp) is composed of platelets 2,3,4,5,6 and 8. Scale bars (**a**–**c**) = 1 µm.

**Figure 5 marinedrugs-21-00216-f005:**
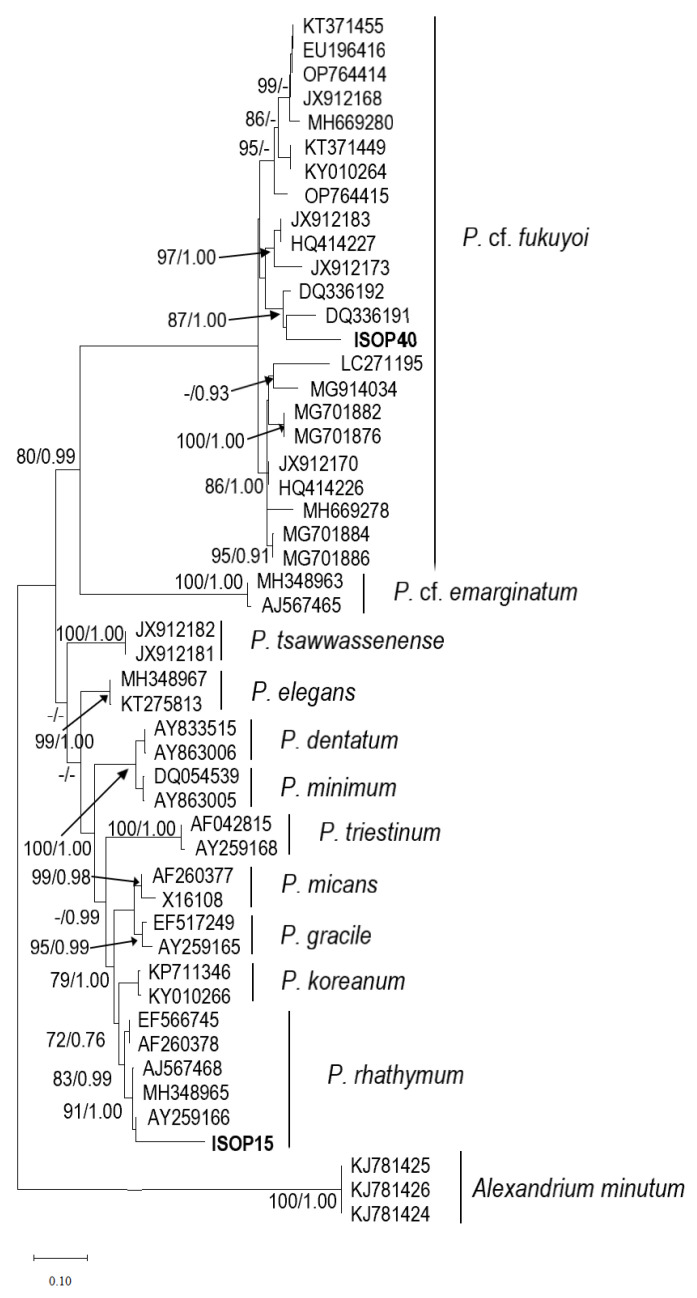
LSU rRNA phylogeny showing the relationships between *Prorocentrum* sequences (ISOP40 & ISOP15) obtained in the present study and other allied species. Numbers at nodes represent the result of the ML bootstrap analysis. ‘-’ represents unsupported value. Accession numbers: ISOP40 (OQ352626); ISOP15 (OQ352630).

**Figure 6 marinedrugs-21-00216-f006:**
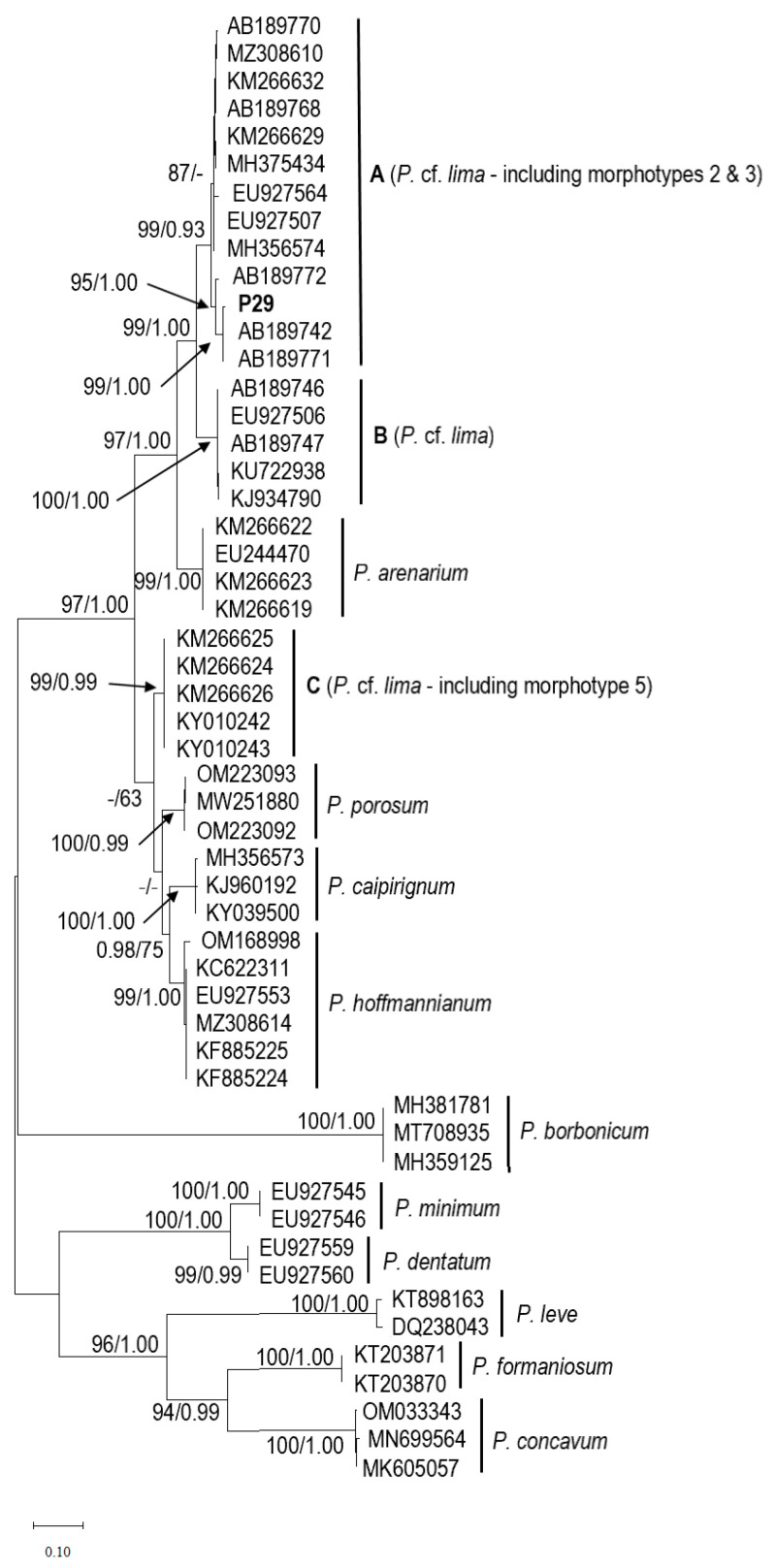
Maximum likelihood phylogeny of *Prorocentrum lima* species complex for the ITS rDNA regions. Bold label (P29) sequence was obtained in this study. Nodal support is Bayesian posterior probability (PP), and bootstrap (BT) values obtained from Bayesian inference analysis and maximum likelihood analysis, shown as ML/BI, respectively. ‘-’ represents unsupported value. Accession number: ISOP29 (OQ352631).

**Figure 7 marinedrugs-21-00216-f007:**
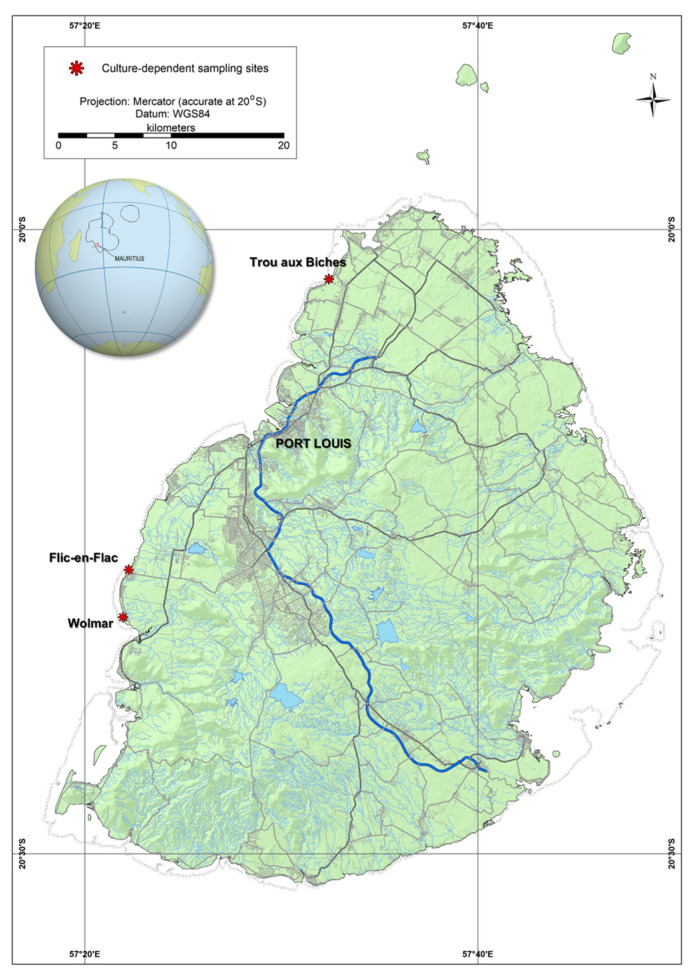
Map showing the three sampling locations, Mauritius.

**Table 1 marinedrugs-21-00216-t001:** Mean zone of inhibition of the three extracts from ISOP15, P29, and ISOP40 against Gram-positive and negative microorganisms.

Isolates	Zone of Inhibition						
	Gram-Positive Strains ^a^			Gram-Negative Strains ^b^			PC
	1	2	3	4	5	6	
Intracellular Extract							
ISOP15	12 ± 0.2	20 ± 0.2	20 ± 0.25	30 ± 0.0	20 ± 0.25	10 ± 0.42	20 ± 0.3
P29	13 ± 1.1	25 ± 0.4	23. ± 0.4	24 ± 0.8	15 ± 0.4	27 ± 0.0	24 ± 1.5
ISOP40	15 ± 0.4	19 ± 0.4	15 ± 0.4	21 ± 0.4	21 ± 0.4	15 ± 0.4	14 ± 1.1
Extracellular Protein							
ISOP15	11 ± 0.2	7 ± 0.5	14 ± 0.2	27 ± 0.5	14 ± 0.2	15 ± 0.4	23 ± 1.0
P29	15 ± 0.4	11 ± 0.0	7 ± 0.5	18 ± 0.5	16 ± 0.3	7 ± 0.4	13 ± 1.1
ISOP40	12 ± 0.5	16 ± 1.2	9 ± 0.4	20 ± 0.4	17 ± 0.5	7 ± 0.5	18 ± 0.5
Extracellular Polysaccharides							
ISOP15	14 ± 0.2	20 ± 0.3	16 ± 1.1	28 ± 0.2	1 4 ± 1.1	15 ± 0.4	19 ± 0.2
P29	14 ± 0.7	13 ± 0.1	12 ± 0.5	21 ± 0.6	15 ± 0.4	14 ± 1.5	24 ± 1.1
ISOP40	24 ± 0.4	15 ± 0.4	15 ± 0.4	15 ± 0.4	15 ± 0.4	11 ± 0.2	20 ± 1.5

^a^ Gram-positive bacteria: 1. MRSA, 2. *E. faecalis*, 3. *S. pneumonia* and ^b^ Gram-negative bacteria: 4. *V. parahaemolyticus*, 5. *P. mirabilis*, 6. *S.typhimurium.* PC: Positive control (Streptomycin—10 µg/mL) and a Negative control (DMSO); values are averages of five replicates, ± depicts standard deviation.

**Table 2 marinedrugs-21-00216-t002:** Minimum inhibitory concentration of the isolates extracts against pathogens.

Isolates	Minimum Inhibitory Concentration (μg/mL)					
	Gram-Positive Strains ^a^			Gram-Negative Strains ^b^		
	1	2	3	4	5	6
Intracellular Extract						
ISOP15	5	2.5	1.25	2.5	2.5	5
P29	5	2.5	0.625	1.25	5	1.25
ISOP40	1.25	1.25	1.25	0.625	5	2.5
Extracellular Protein						
ISOP15	10	10	5	0.625	2.5	5
P29	5	5	10	5	5	10
ISOP40	2.5	5	5	2.5	2.5	10
Extracellular Polysaccharides						
ISOP15	1.25	5	2.5	0.625	5	5
P29	5	5	5	2.5	5	2.5
ISOP40	0.625	2.5	5	5	5	2.5
Positive Control						
10 µg/mL streptomycin	0.625	1.25	1.25	0.625	0.625	0.625

Initial concentration of extracts: 10 µg/mL. ^a^ Gram-positive bacteria: 1. MRSA, 2. *E. faecalis*, 3. *S. pneumonia* and ^b^ Gram-negative bacteria: 4. *V. parahaemolyticus*, 5. *P. mirabilis*, 6. *S. typhimurium*. PC: Positive control (Streptomycin initial concentration 10 µg/mL) and a Negative control (DMSO).

## Data Availability

The sequence data presented in this study are openly available in Genbank.

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
