# Peer review of "Ribosomal DNA Sequence-Based Taxonomy and Antimicrobial Activity of Prorocentrum spp. (Dinophyceae) from Mauritius Coastal Waters, South-West Indian Ocean"

_marinedrugs, 2023, doi:10.3390/md21040216_

Round 1
Reviewer 1 Report
Musai et all submitted manuscript regarding Ribosomal DNA sequence-based taxonomy and antimicrobial 2 activity of Prorocentrum spp. (Dinophyceae) from Mauritius 3 coastal waters, South-West Indian Ocean. I have several comments:
1. The importance of the Ribosomal DNA sequence-based taxonomy was not well described and thus the introduction was not fit very well with the title of the manuscript. The connection of the Ribosomal DNA sequence-based taxonomy within the manuscript was not found
2. Need one sentence of the introduction (at the beginning) in the abstract
3. The novelty of the the manuscript was not well explain
4. The diameter of zone inhibition from negative control were not shown in Table 1 and 2.
5. The inhibition of the positive control was not mention in table 2 as well
Conclusion, the manuscript was not fit very well with the marine drug scope and more suitable for the molecular microbiology and biology journals
Author Response
Please see attachment
|
Reviewer 1 |
|||
|
Number |
Line |
Comment |
Corrections/Authors comments |
|
|
|
The importance of the Ribosomal DNA sequence-based taxonomy was not well described and thus the introduction was not fit very well with the title of the manuscript. The connection of the Ribosomal DNA sequence-based taxonomy within the manuscript was not found |
This has been addressed in the discussion and we have also cited a paper we recently published using ribosomal DNA as well across species of microalgae. The connecttion
Added in introduction: Traditional methods to assess microalgal diversity from coastal environment relied mostly on traditional microscopy methods. However given the problems arising from phenotypic plasticity and cultural variation, there has been a need to incorporate use of DNA sequence data. The latter, especially ribosomal DNA, has been widely used, not only across microalgae but among a wide range of microorganisms |
|
|
|
Need one sentence of the introduction (at the beginning) in the abstract |
Added one sentence in the introduction |
|
|
|
The novelty of the manuscript was not well explain |
The novelty is that this is first species being isolated and characterised based on sequence data in this region. In addition its antimicrobial properties are also elucidated which brings forward novelty in our findings as well. |
|
|
|
The diameter of zone inhibition from negative control were not shown in Table 1 and 2. |
Included sentence: “While the negative control produced no observable zones.” |
|
|
|
The inhibition of the positive control was not mention in table 2 as well |
The table has been revised to include the information. |
Reviewer 2 Report
Please add accession numbers: OQ352631; 297 OQ352626; OQ352630 to your phylogenetic trees
Lines 242-249. Need to add more references.
Authors should follow the comments of reviewer. See attached file please.

Author Response
|
Reviewer 2 |
|||
|
Number |
Line |
Comment |
Corrections/Authors comments |
|
|
Line 32 |
of what?? Potentially of interest for the pharmaceutical industry |
We corrected this: From: “with potential for application in a number of traditional and emerging fields”
To: “of potential interest to the pharmaceutical industry” |
|
|
Lines 54,56,58,59, 61-67 |
Figs 1a, c, f is not providing taxonomic information Nucleus position should be indicated, where is it in Fig. 1b |
Lines: 54, 56,58, 59
Figure 1: updated.
Caption: Figure 1 |
|
|
|
Figure 2a-b is not clear, did not clearly show species characters. Figure must be approved |
Updated
Unfortunately, that the only micrographs we are having.
|
|
|
|
Notes in figure is very small, unclear. Need to approve |
Figure 3: updated. |
|
|
Line 102 |
Figure4: Scale bars should be bigger, 1µm is very small for cell size (appr. 30 µm) |
Scale bars are based on the SEM micrographs. Original micrographs are herein provided. |
|
|
Line 126 |
Figure 5: GB access number shoul be add after strain name, ISOP40 |
GB numbers included in caption |
|
|
Line 142 |
Figure 6: Same as above |
GB number included in caption |
|
|
Line 242-249 |
THIS APPEARS TO BE THE MOST IMPORTANT CONTRIBUTION OF THIS MS – BUT WITHOUT REFERENCES ??!! THERE ARE PRESUMABLY FEW STUDIES FOR COMPARISON – IF ANY, THESE STUDIES SHOULD BE INCLUDED HERE AND IN CASE THERE ARE NO PREVIOUS STUDIES WHICH ARE RELEVANT TO CITE THERE, THIS SHOULD BE MENTIONED |
This has been revised. We have included a paragraph to substantiate this. The paragraph has been highlighted in the discussion. |
|
|
Line 297 |
Pls add the strain number after acces. number, eg OQ352631 (strain P29), so on with others |
Included strain numbers |
|
|
Line 385 |
No graphs were found in result section |
From: “…graphs were generated using…”
To: “…tabulated…” |
Round 2
Reviewer 1 Report
The manuscript has been well written and deserved for publication. As the antimicrobial activity was very good, it would be better to know that the antimicrobial activity reported here are not due to cytotoxic property of the extracts.